

# Speed of sound in dense strong-interaction matter

Jens Braun[1,2], Andreas Geißel[1] and Benedikt Schallmo[1]

**1** Institut für Kernphysik, Technische Universität Darmstadt, D-64289 Darmstadt, Germany
**2** ExtreMe Matter Institute EMMI, GSI, Planckstraße 1, D-64291 Darmstadt, Germany

## Abstract

We study the speed of sound in strong-interaction matter at zero temperature and in density regimes which are expected to be governed by the presence of a color-superconducting gap. At (very) high densities, our analysis indicates that the speed of sound approaches its asymptotic value associated with the non-interacting quark gas from below, in agreement with first-principles studies which do not take the presence of a color-superconducting gap into account. Towards lower densities, however, the presence of a gap induces an increase of the speed of sound above its asymptotic value. Importantly, even if gap-induced corrections to the pressure may appear small, we find that derivatives of the gap with respect to the chemical potential can still be sizeable and lead to a qualitative change of the density dependence of the speed of sound. Taking into account constraints on the density dependence of the speed of sound at low densities, our general considerations suggest the existence of a maximum in the speed of sound. Interestingly, we also observe that specific properties of the gap can be related to characteristic properties of the speed of sound which are indirectly constrained by observations.

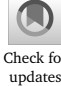

# 1 Introduction

The impressive progress made in the observation of neutron-star mergers via gravitational-wave signals [1, 2] together with the advances made in direct measurements of the radius and the mass of heavy neutron stars [3–12] challenges our understanding of the properties of dense strong-interaction matter. For example, constraints from neutron-star masses suggest the existence of a maximum in the speed of sound in dense matter which exceeds the asymptotic value of a non-interacting quarks gas [13–19], preferably at densities $n/n_0 \lesssim 10$ (where $n_0$ is the nuclear saturation density).

In the present work, we shall only consider the zero-temperature limit where the speed of sound $c_s$ can be written as a ratio of derivatives of the pressure $P$ with respect to the chemical potential $\mu$:

$$c_s = \frac{1}{\sqrt{\mu}} \left( \frac{\partial P}{\partial \mu} \right)^{\frac{1}{2}} \left( \frac{\partial^2 P}{\partial \mu \partial \mu} \right)^{-\frac{1}{2}} . \tag{1}$$

From this equation, it becomes apparent that this quantity is a sensitive probe for the analysis of the density dependence of the pressure of strong-interaction matter. Indeed, this expression suggests that seemingly small contributions to the equation of state may already lead to significant changes in the density dependence of the speed of sound, depending on the scaling of these contributions with the density. Therefore, already a qualitatively correct description of the density dependence of the speed of sound in strong-interaction matter requires a detailed understanding of the relevant degrees of freedom at work at different densities and their dynamics.

In the low-density regime, where the dynamics is governed by spontaneous chiral symmetry breaking with nucleons and pions as effective degrees of freedom, chiral effective field theory (EFT) provides a framework to describe nuclear matter in a systematic fashion [20]. In particular, it opens up the opportunity to constrain properties of nuclear matter at low densities [21]. Specifically for the speed of sound, studies based on chiral EFT predict a rapid increase with the density [22, 23].

Considering the high-density regime, we first note that the chiral symmetry of the theory of the strong interaction (Quantum Chromodynamics, QCD) is expected to be at least effectively restored. However, the ground state is still nontrivial. In fact, already early ground-breaking studies of the theory of the strong interaction, ranging from low-energy model studies [24–27] to first-principles studies in the weak-coupling limit [28–34], pointed out that strong-interaction matter at sufficiently low temperatures and high densities is a color superconductor, due to the presence of a Bardeen-Cooper-Schrieffer-type instability, see Refs. [35–46] for reviews. In recent studies based on the functional renormalization group (fRG) approach, it has then been found that the presence of a color-superconducting gap in the excitation spectrum of the quarks gives rise to a maximum in the speed of sound [23, 47, 48]. Notably, in accordance with constraints from nuclear physics and observations [13–19], this maximum exceeds the asymptotic value associated with the non-interacting quark gas, for both isospin-symmetric matter and neutron-star matter.

Finally, at very high densities and under the assumption that the color-superconducting gap does not contribute significantly to the equation of state, perturbative calculations suggest that the speed of sound eventually approaches its asymptotic value from below [49–58], see also Ref. [23] for a discussion.

With the present work, we aim at an analysis of the density dependence of the speed of sound and the identification of mechanisms underlying qualitatively different scenarios. In particular, our analysis allows to relate the size of the color-superconducting gap to the specific value of the density at which the speed of sound exceeds its asymptotic value when the density is decreased starting from asymptotically high densities. To this end, we first discuss the form

of the equation of state in the presence of a color-superconducting gap in Sec. 2. In Sec. 3, we then analyze the speed of sound in detail for the case of two massless quark flavors coming in three colors. A generalization of our considerations to the phenomenologically most relevant case of $(2+1)$ flavors is in principle possible but is beyond the scope of this work. Nevertheless, we believe that our present study already provides valuable information on general properties of dense strong-interaction matter. A brief discussion of this aspect can be found in Sec. 4 together with our conclusions.

## 2 Equation of state

Throughout this work, we shall restrict ourselves to the isospin-symmetric limit at zero temperature in density regimes which are governed by the presence of a color-superconducting gap.

Our analysis of the density dependence of the speed of sound in Sec. 3 builds on an expansion of the equation of state in the presence of a color-superconducting gap. In this section, we therefore discuss this expansion on general grounds. To be more specific, in Subsec. 2.1, we first consider the expansion of the pressure in the case where the gauge coupling is treated as a fixed "external" parameter. In Subsec. 2.2, we then give a brief discussion in the context of fully non-perturbative calculations.

### 2.1 Expansion of the equation of state

Let us start our discussion by considering the pressure in the non-interacting limit:

$$P = P_{\text{SB}} = \frac{\mu^4}{2\pi^2},\tag{2}$$

where $\mu$ is the quark chemical potential. Turning on the strong coupling $g$, a color-superconducting gap $|\Delta_0|$ is generated in the excitation spectrum of the quarks, even for infinitesimally small values of $g$ because of a Bardeen-Cooper-Schrieffer-type instability in the system (see, e.g., Refs. [36–41, 59] for detailed discussions of this aspect). Since the strong coupling is dimensionless and we assume it to be a constant parameter for the time being, the chemical potential is the only scale in the problem. Thus, we have $|\Delta_0| = |\Delta_0(\mu, g)| = \mu f_\Delta(g)$. The dimensionless function $f_\Delta(g)$ depends only on the coupling $g$.

In the weak-coupling limit at high densities, the gap can be computed analytically. For example, for the chirally symmetric gap (with $J^P = 0^+$) associated with pairing of the two-flavor color-superconductor (2SC) type, it was found that [28–31, 34]

$$|\Delta_0| \sim \mu g^{-5} \exp\left(-\frac{3\pi^2}{\sqrt{2}g}\right).\tag{3}$$

With respect to the dependence of the gap to the coupling, it should be added that Bardeen-Cooper-Schrieffer-type gaps in the fermion spectrum are in general expected to be non-analytic smooth functions of the coupling $g$. In particular, an approximation of the gap in terms of a Taylor series about $g = 0$ does not exist, see, e.g., Refs. [39, 41] for reviews. Note also that the gap is directly related to the expectation value of a quark bilinear. For the gap in Eq. (3), for example, we have

$$\Delta_0^a \sim \langle \psi_b^T \mathcal{C} \gamma_5 \tau_2 \epsilon_{abc} \psi_c \rangle,\tag{4}$$

where $\mathcal{C}$ is the charge-conjugation operator, $\tau_2$ is the second Pauli matrix, and, in color space, it is summed over the totally antisymmetric tensor $\epsilon_{abc}$. Note that $|\Delta_0|^2 := \sum_a |\Delta_0^a|^2$ is a gauge-invariant quantity.

The gap as given in Eq. (3) only exhibits a trivial dependence on the chemical potential which arises from the fact that $\mu$ is the only dimensionful quantity if the coupling $g$ is treated as a constant "external" parameter. In a non-perturbative computation which takes the scale dependence of the coupling into account, however, the gap may acquire a non-trivial dependence on the chemical potential $\mu$ since the chemical potential then has to be measured in units of the scale set by the running of the coupling, which is $\Lambda_{\text{QCD}}$. The latter scale is also present in the vacuum limit. We shall come back to this below but focus on a constant coupling $g$ for the moment.

In a computation of the pressure (which is essentially given by the quantum effective action $\Gamma$ evaluated at its minimum), one has to take into account that the quark and gluon propagators are potentially altered in the presence of a gap. However, not all gluons and quarks are directly affected by the gap. For example, because of the underlying Anderson-Higgs-type mechanism [60–64] associated with the symmetry-breaking pattern SU(3) → SU(2) in color space, only five of the eight gluons effectively acquire an effective mass $\sim \Delta_0$, see, e.g., Refs. [39, 41] for a review. This suggests that the pressure is a function of the coupling $g$ and the gap, $P = P(g, |\Delta_0|^2)$. Employing now dimensional and symmetry arguments, we arrive at the following expansion of the pressure in terms of the dimensionless gauge-invariant quantity $|\bar{\Delta}_0|^2 = |\Delta_0/\mu|^2$:[1]

$$P = P_{\text{SB}} \left( \gamma_0(g) + \gamma_1(g)|\bar{\Delta}_0|^2 + \frac{1}{2}\gamma_2(g)|\bar{\Delta}_0|^4 + \dots \right), \tag{5}$$

where $\Delta_0$ is assumed to be homogeneous and

$$\gamma_i(g) = \frac{\mu^{2i}}{P_{\text{SB}}} \frac{\partial^i P(g, |\Delta_0|^2)}{(\partial |\Delta_0|^2)^i} \Bigg|_{|\Delta_0|=0}. \tag{6}$$

The dependence on the chemical potential is fully determined by the non-interacting quark gas since the coupling is still considered to be a constant and $\mu$-independent parameter. Note that we assume that the pressure is an analytic function of the gap which should be the case away from a phase transition.

The first term in the expansion (5) is the pressure in the absence of a gap. Thus, we have

$$\gamma_0(g) = 1 + \mathcal{O}(g^2). \tag{7}$$

The $g$-dependent corrections can be extracted from perturbative calculations [49–58], provided that the gauge coupling is sufficiently small.

Since the pressure is related to the effective action, the functions $\gamma_i$ can be extracted from correlation functions evaluated at vanishing gap. For example, $\gamma_1$ can be related to the diquark propagator and therefore to a four-quark correlation function of the following form:

$$\left\langle (\bar{\psi}_b \tau_2 \epsilon_{abc} \gamma_5 \mathcal{C} \bar{\psi}_c^T)(\psi_d^T \mathcal{C} \gamma_5 \tau_2 \epsilon_{ade} \psi_e) \right\rangle \Big|_{|\Delta_0|=0}. \tag{8}$$

For example, non-perturbative methods may be employed to compute $\gamma_1$, see, e.g., Ref. [47]. From this study, we deduce that

$$\gamma_1(g) = 2 + \mathcal{O}(g^2). \tag{9}$$

---

[1]Here, we exploit the fact that we can at least formally compute the quantum effective action $\Gamma$ in the presence of an auxiliary field (e.g., associated with a quark bilinear). In the underlying path integral, this requires to introduce a suitably chosen source term. In any case, for our present purposes, one has to choose an auxiliary field which agrees identically with the gap at the minimum of the corresponding effective action. The effective action can then be written as a power series of this auxiliary field. The functions $\gamma_i$ are therefore related to correlation functions, see below. Note that the pressure and the effective action are related: $P = -\Gamma_0/V_4$, where $\Gamma_0$ is the effective action evaluated at its physical minimum and $V_4$ is the spacetime volume.

Note that we expand the pressure in powers of $|\Delta_0|^2$ and not in the condensate. Therefore, $\gamma_1$ is finite for $g \to 0$. For $|\Delta_0|^2$, however, we have $|\Delta_0|^2 \to 0$ for $g \to 0$.

A computation of $g$-dependent corrections to the function $\gamma_1(g)$ is beyond the scope of the present work. However, we can make a general statement about this function by exploiting the fact that the ground state of strong-interaction matter is expected to be a color superconductor at sufficiently high densities. In fact, this implies that the pressure in this density regime should be greater than the pressure in the absence of a gap. If this was not the case at some (high) density, then the system would undergo a phase transition to an ungapped phase (as the ground state is associated with the phase with lowest Gibbs energy, i.e., highest pressure). Therefore, we conclude that $\gamma_1(g) > 0$ for sufficiently high densities, where $\bar{\Delta}_0$ is small and therefore terms of the order $\sim \bar{\Delta}_0^4$ and higher can be dropped in Eq. (5).

In general, the coefficient functions $\gamma_j$ in Eq. (5) are associated with correlation functions of $4j$ quarks. For example, eight-quark correlation functions are required to compute the function $\gamma_2(g)$. With respect to the relevance of terms with $j > 1$, we note that such terms have been observed to be subleading over a wide density range in a non-perturbative computation of the speed of sound [47]. Of course, the relevance of these terms ultimately depends on the density and the details of the gap (such as its size and density dependence).

We would like to add that, for $\gamma_0 = 1$, $\gamma_1 = 2$ and $\gamma_i = 0$ ($i > 1$), we recover the approximation of the pressure which has already been used in early studies of dense strong-interaction matter, see, e.g., Refs. [36, 65, 66]. Generally speaking, the $g$-independent contributions to the $\gamma_i$-functions can be extracted from a one-loop approximation of the effective action of QCD which only takes the quark loop in the presence of a gap into account, see Refs. [47, 67] for a discussion in the context of renormalization-group studies. Terms depending on the coupling $g$ are generated by, e.g., quantum corrections to the gluon polarization tensor. However, we emphasize that a computation of the $\gamma_i$-functions does not necessarily require to specify the functional form of the gap. Indeed, for the expansion (5), we have only assumed the existence of a gap.

Let us finally comment on the dependence of the expansion (5) on the chemical potential. Up to this point, the chemical potential is the only dimensionful scale in our analysis since we have assumed that the coupling $g$ is a constant parameter. Therefore, the gap must be proportional to $\mu$ and the pressure (5) must be proportional to $\mu^4$, i.e., we have $P/P_{\mathrm{SB}} = f_P(g)$ with a dimensionless function $f_P$ depending only on $g$ but not on $\mu$. A non-trivial dependence on the chemical potential can be introduced by taking into account that the coupling carries an implicit dependence on the chemical potential. For example, this may be estimated by evaluating the coupling in a one-loop approximation at the chemical potential $\mu$: $g^2(\mu/\Lambda_{\mathrm{QCD}}) = 1/(b_0 \ln(\mu/\Lambda_{\mathrm{QCD}}))$. From a phenomenological standpoint, this corresponds to assuming that the typical momentum transfer in interaction processes is of the order of the chemical potential $\mu$. In any case, by using $g^2(\mu/\Lambda_{\mathrm{QCD}})$ in Eq. (5), we effectively replace the "parameter" $g$ with the dimensionless quantity $\mu/\Lambda_{\mathrm{QCD}}$:

$$P = P_{\mathrm{SB}} \left( \gamma_0(g(\mu/\Lambda_{\mathrm{QCD}})) + \gamma_1(g(\mu/\Lambda_{\mathrm{QCD}}))|\bar{\Delta}_0|^2 + \dots \right), \tag{10}$$

where $\bar{\Delta}_0 = \bar{\Delta}_0(\mu, g(\mu/\Lambda_{\mathrm{QCD}}))$. In this way, the pressure acquires a non-trivial dependence on the chemical potential. With respect to a computation of the speed of sound $c_{\mathrm{s}}$, we note that the dependence on $\mu/\Lambda_{\mathrm{QCD}}$ is essential. In fact, we only have $c_{\mathrm{s}} = 1/\sqrt{3}$ (i.e., the value of the non-interacting quark gas), if the coupling $g$ is assumed to be independent of the chemical potential.

## 2.2 Expansion of the equation of state and non-perturbative approaches

In a fully non-perturbative study, the scale dependence of the coupling is explicitly taken into account in the computation of correlation functions. The scale in such a study is set by fix-

ing the value of the strong coupling at a given scale which can then be translated into the scale $\Lambda_{\text{QCD}}$. At finite chemical potential, this implies that corrections to the equation of state of the non-interacting quark gas in general depend on $\mu/\Lambda_{\text{QCD}}$. For the gap, which can be computed as the expectation value of a quark bilinear, we therefore have

$$\Delta_0^a = \Delta_0^a(\mu, \mu/\Lambda_{\text{QCD}}) \sim \langle \psi_b^T \mathcal{C} \gamma_5 \tau_2 \epsilon_{abc} \psi_c \rangle. \tag{11}$$

The functional form of $|\Delta_0|^2 = \sum_a |\Delta_0^a|^2$ is in general non-trivial and, at lower densities, it may even deviate from the one resulting from Eq. (3) with the coupling $g$ replaced by $g(\mu/\Lambda_{\text{QCD}})$, see, e.g., Refs. [23, 26, 28, 30, 31, 39, 41, 47, 68–70] for corresponding discussions. Indeed, towards lower densities, strong-interaction matter is effectively probed at smaller and smaller momentum scales and therefore corrections beyond the weak-coupling limit may become relevant.

With the gap at hand for a given value of the chemical potential, we can formally write the pressure again as a power series in the gauge-invariant quantity $|\bar{\Delta}_0|^2$:

$$P = P_{\text{SB}}\big(\tilde{\gamma}_0(\mu/\Lambda_{\text{QCD}}) + \tilde{\gamma}_1(\mu/\Lambda_{\text{QCD}})|\bar{\Delta}_0|^2 + \dots\big), \tag{12}$$

which corresponds to Eq. (10). Again, since the ground state is expected to be a color superconductor, we have $\tilde{\gamma}_1 > 0$, at least at sufficiently high densities where higher orders in $|\bar{\Delta}_0|$ can be dropped, see our discussion in the previous subsection. Of course, in a fully non-perturbative study, such an expansion may be of limited interest since the pressure may be available numerically as a function of the chemical potential $\mu$. Still, our considerations can be useful to analyze properties of dense strong-interaction matter, as we shall see next.

## 3 Speed of sound

We now employ the expansion (10) for a *qualitative* analysis of the density dependence of the speed of sound. Throughout this section, we shall set $\gamma_i = 0$ for $i > 1$. This leaves us with

$$P \approx P_{\text{SB}}\big(\gamma_0 + \gamma_1|\bar{\Delta}_0|^2\big). \tag{13}$$

For our qualitative study, we expect that this is sufficient. Indeed, it has been observed in Ref. [47] that terms of order $\sim |\bar{\Delta}_0|^4$ and higher do not alter the qualitative behavior of the speed of sound over a wide density range.

Let us now start by considering the case where the $\gamma_i$-functions are assumed to be independent of $g$. To be specific, we use $\gamma_0(g) = 1$ and $\gamma_1(g) = 2$ as discussed in the previous section. Assuming that $|\Delta_0|/\mu \to 0$ for $\mu \to \infty$, it has been pointed out in Ref. [48] that the speed of sound approaches its asymptotic value from above for $\mu \to \infty$ (i.e., in the limit of infinite density $n$). Note that these assumptions about the gap are consistent with the expected behavior of the gap as a function of the chemical potential, at least for (very) large chemical potentials, see, e.g., Refs. [23, 26, 28, 30, 31, 36–41, 47, 68, 69]. In any case, away from the high-density regime, it has been found before that the gap induces an increase of the speed of sound above the value associated with the non-interacting quark gas [23, 47, 48].

Next, we consider the case with $\gamma_i = 0$ (for $i > 0$), i.e., all gap-induced corrections to the pressure are dropped. For $\gamma_0$, we choose

$$\gamma_0(g) = 1 - \frac{g^2}{2\pi^2} + \mathcal{O}(g^3). \tag{14}$$

This leads us to the perturbative result for the pressure at leading order in the coupling $g$, see Refs. [49–52]. For the coupling, we now employ the standard one-loop result evaluated at the

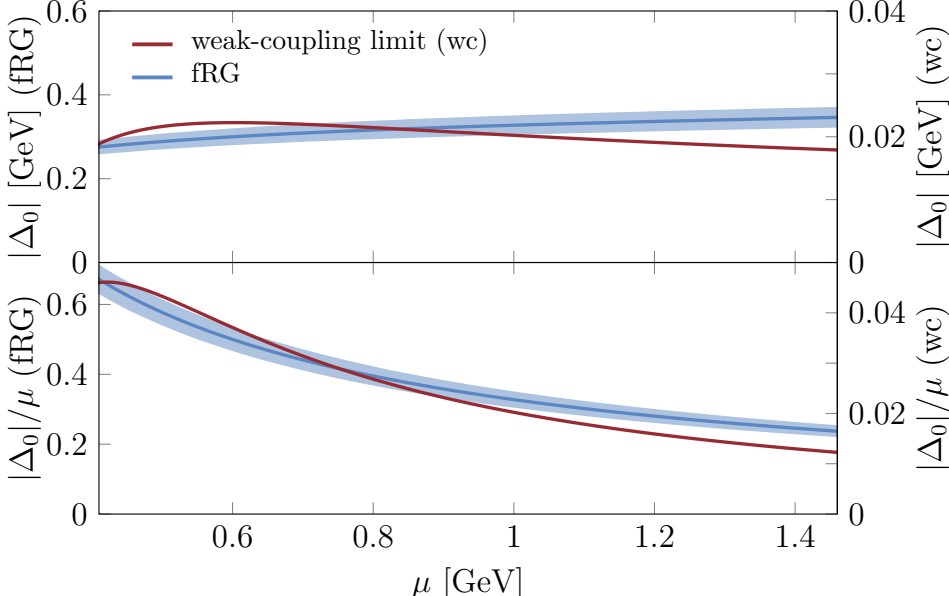

Figure 1: Color-superconducting $|\Delta_0|$ (top panel) and $|\Delta_0|/\mu$ (bottom panel) as a function of the chemical potential $\mu$ as obtained from a study in the weak-coupling limit [28, 30, 31, 39], see Eq. (16), and a recent fRG study at low and intermediate densities [47]. The (blue) band is associated with the fRG data and results from a variation of the regularization scheme and the experimental uncertainty in the strong coupling.

scale set by the chemical potential:[2]

$$g^2(\mu/\Lambda_{\text{QCD}}) = \frac{1}{b_0 \ln(\mu/\Lambda_{\text{QCD}})}. \tag{15}$$

Here, $\Lambda_{\text{QCD}} = \Lambda_0 \exp(-1/(b_0 g_0^2))$ with $b_0 = 29/(24\pi^2)$ and $g_0$ is the value of the strong coupling at the scale $\Lambda_0$.[3] Using now Eq. (1), we find that the speed of sound is smaller than its asymptotic value for all densities considered in this work. Moreover, we observe that the speed of sound approaches its asymptotic value from below for $n \to \infty$, see also our discussion below. Note that this remains unchanged, even if higher-order corrections in $\gamma_0$ are taken into account [23, 56–58].

For small $|\Delta_0|/\mu$, it may be tempting to drop corrections associated with the gap in the expansion (10), such that the pressure is given by the pressure of the ungapped system (as described by $\gamma_0$). In fact, the gap-induced corrections vanish identically for $\mu \to \infty$, provided that $\Delta_0/\mu \to 0$ for $\mu \to \infty$. At least at first glance, it may therefore be reasonable to drop gap-induced corrections in a computation of the pressure. For the speed of sound, however, the situation may be different since it is essentially given by the ratio of the first and second derivative of the pressure with respect to the chemical potential, see Eq. (1). Even if the gap-induced terms to the pressure may appear small, their derivatives may still yield sizeable contributions to the speed of sound. To analyze this aspect, we choose $\gamma_0$ as given in Eq. (14) and $\gamma_1 = 2$. This corresponds to combining the two cases discussed above. Moreover, we now

---

[2]Of course, this is not fully consistent since the correction $\sim g^2$ in Eq. (14) is generated by a two-loop diagram. For our qualitative analysis of the speed of sound, however, this is of no relevance.

[3]In our numerical calculations, we choose $g_0^2/(4\pi) \approx 0.179$ and $\Lambda_0 = 10\,\text{GeV}$ [71]. This yields $\Lambda_{\text{QCD}} \approx 0.265\,\text{GeV}$. In order to avoid that our analysis is spoilt by the Landau pole associated with the scale $\Lambda_{\text{QCD}}$, we ensure that the chemical potential is (sufficiently) greater than the scale $\Lambda_{\text{QCD}}$ in our computations.

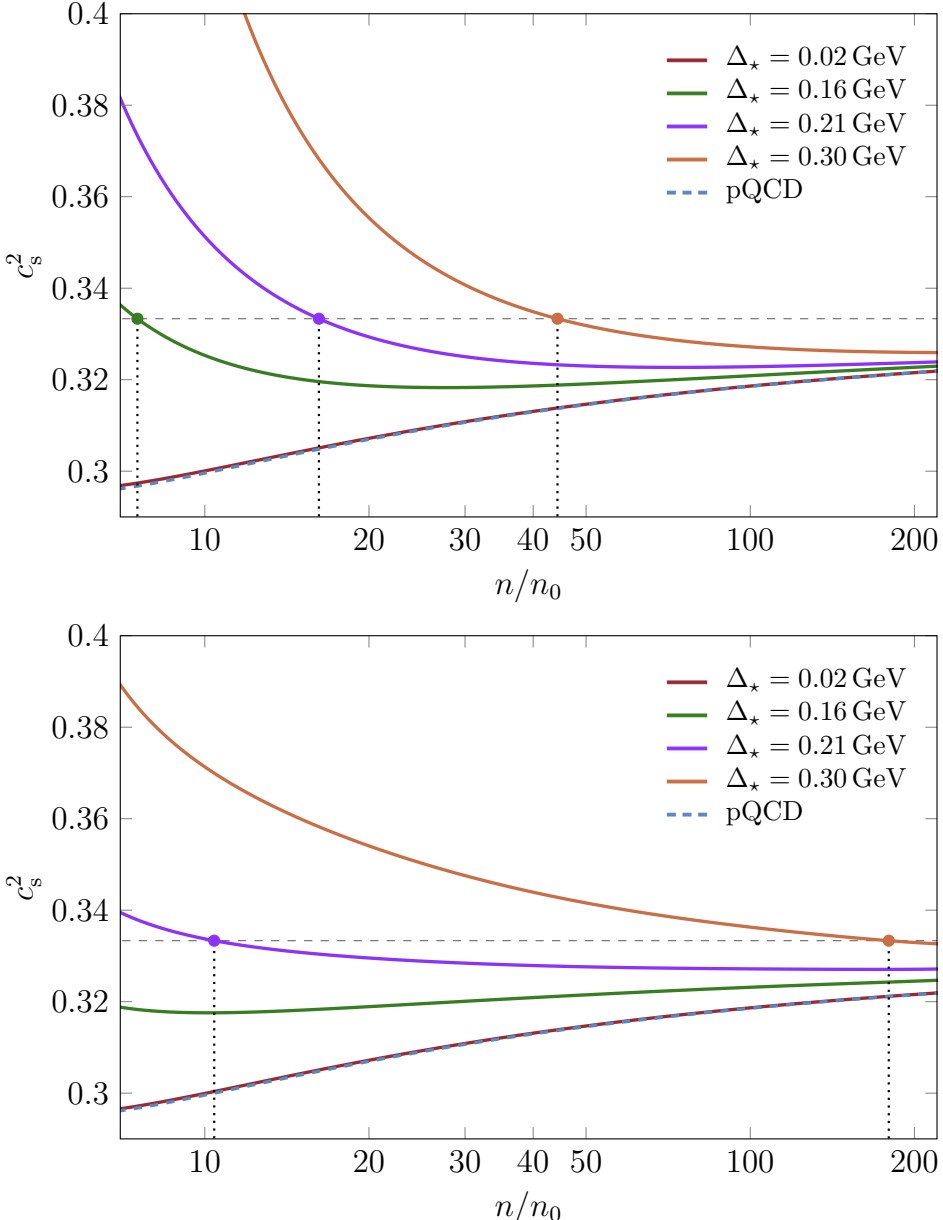

Figure 2: Speed of sound (squared) as a function of the baryon density $n$ in units of the nuclear saturation density $n_0$ for different values $\Delta_\star$ of the gap at $n/n_0 = 10$, $\Delta_\star = |\Delta_0(n/n_0 = 10)|$. For comparison, it has been found that $|\Delta_0| \approx 0.07 \ldots 0.16$ GeV at $n/n_0 \approx 5$ in an early low-energy model study [24]. Note that the gap in this type of model studies increases with increasing densities. The dashed horizontal line corresponds to the speed of sound squared of the non-interacting quark gas. The blue dashed line (perturbative QCD, pQCD) is the speed of sound as obtained by choosing $\gamma_0$ as given in Eq. (14) and setting $\gamma_i = 0$ for $i > 0$. Top panel: Speed of sound (squared) as obtained by using a gap with a functional form as found in the weak-coupling limit, see Eq. (16). The parameter $s_0$ has been tuned such that $\Delta_\star = 0.02$ GeV, 0.16 GeV, 0.21 GeV, 0.30 GeV ($\Delta_\star = 0.02$ GeV corresponds to $s_0 = 1$). Bottom panel: Speed of sound (squared) as obtained by employing a gap with a functional form as found in a recent fRG calculation [47]. To obtain the different values for $\Delta_\star$, we have simply rescaled the gap, as also done for the gap in the weak-coupling limit.

have to specify the functional form of the gap in Eq. (13). Since the precise functional form of the gap over a wide density range is still unknown, we employ the results for the gap from two different calculations: the gap from a calculation in the weak-coupling limit [30, 31, 39] and the gap from a recent fRG study at low and intermediate densities [47], where strong-interaction matter is expected to enter the strong-coupling regime. With these results for the gap at hand, we can then gain a better understanding of how the size of the gap and its density dependence affects the speed of sound.

In the weak-coupling limit, the gap can be computed analytically [30, 31, 39]:

$$|\Delta_0| = s\mu g^{-5} \exp\left(-\frac{3\pi^2}{\sqrt{2}g}\right), \qquad (16)$$

where $s = 512\pi^4 \exp(-(4+\pi^2)/8)s_0$. Here, we have introduced a dimensionless parameter $s_0$ which allows us to vary the size of the gap. For $s_0 = 1$, we recover the gap found in Refs. [30, 31, 39]. Note that a variation of $s_0$ only slightly affects the density dependence of the gap.

The results for the gap from the aforementioned fRG study are only available in numerical form [47]. In Fig. 1, we show the fRG results for the gap as a function of the chemical potential together with the results in the weak-coupling limit.[4] Note that, at low densities, the gaps found in recent fRG studies are consistent with those found in conventional low-energy model studies [23, 24, 47]. Still, in our computations below, we shall also vary the size of the gap computed in Ref. [47] by simply rescaling it with a constant prefactor in order to analyze how the size of the gap affects the speed of sound in this case.

In Fig. 2, we present our results for the speed of sound (squared) as a function of the baryon density $n = (\partial P/\partial\mu)/3$ as obtained from choosing $\gamma_0$ as given in Eq. (14) and $\gamma_1 = 2$. The results from a calculation with the gap (16) is shown in the top panel of this figure whereas the results from a calculation based on the gap from a recent fRG study can be found in the bottom panel. We observe that the qualitative behavior of the speed of sound as a function of the density is the same in both cases. Indeed, the speed of sound approaches its asymptotic value associated with the non-interacting quark gas from below for $n \to \infty$. Moreover, starting at (very) high densities, we find that the speed of sound first decreases in both cases when the density is lowered and remains close to the speed of sound as obtained from a computation in the absence of a gap (associated with $\gamma_2 = 0$). Importantly, we also observe that the effect of a color-superconducting gap on the speed of sound becomes continuously stronger with decreasing density and eventually leads to the emergence of a local minimum in the speed of sound at $n = n_{\min}$.

---

[4]From Fig. 1 we deduce that the gap obtained in the weak-coupling limit, see Eq. (16), and the one obtained from the fRG study presented in Ref. [47] do not agree as a function of the chemical potential. This difference can be traced back to the mechanisms which underly the generation of the gap in the weak-coupling calculation and the ones which are at work in the fRG study. To be specific, the weak-coupling calculation relies on the assumptions that the coupling is a constant parameter and that the strength of the coupling is small such that the formation of the gap is not triggered by gluon-induced quark interactions. The presence of a Cooper instability and the associated BCS (Bardeen-Cooper-Schrieffer) mechanism are therefore essential and underlie the formation of the gap in this case. However, the assumptions entering the weak-coupling calculation are effectively only realized at (very) high densities, i.e., for $\mu \gg \Lambda_{QCD}$. Here, $\Lambda_{QCD}$ only serves as a rough estimate for the scale at which the gauge coupling becomes strong. In the fRG study presented in Ref. [47], the dynamics is driven by the fact that, especially towards the lower end of the density range considered in this work, the gauge coupling can already become strong enough in the RG flow to trigger the formation of a gap in the quark excitation spectrum by itself. This is similar to the case of chiral symmetry breaking at zero density which is trigged by the gauge coupling becoming sufficiently strong, see, e.g., Ref. [72] for a review. Of course, because of the Cooper instability, the BCS mechanism is also present in the fRG study but is not the only driving "force" over a significant density range. A detailed discussion of the competition of these two mechanisms for the generation of the gap and the consequences for its density dependence will be presented elsewhere. Note that our analytic study of the density dependence of the speed of sound presented below does not rely on these details underlying the scaling behavior of the gap.

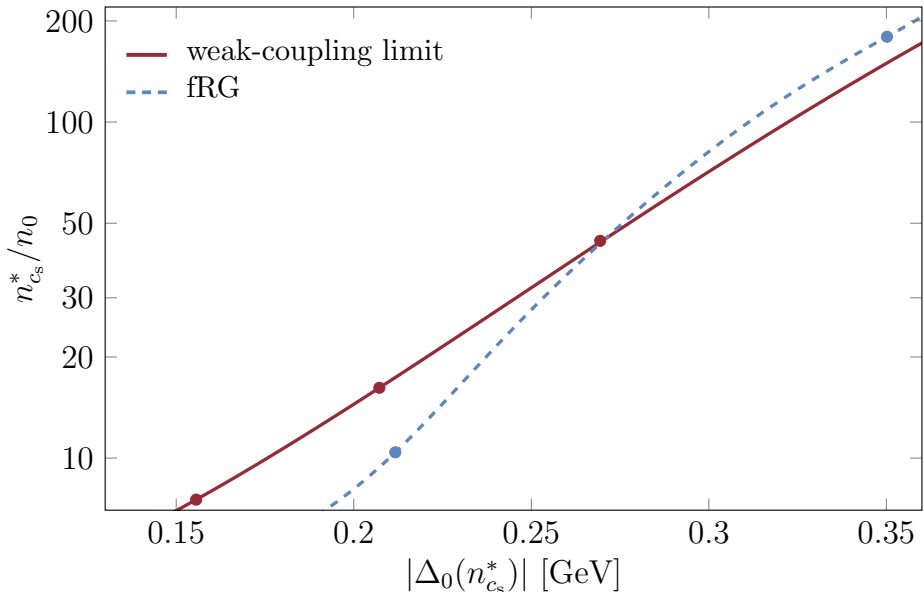

Figure 3: Crossing density $n^*_{c_s}$ (density at which the speed of sound crosses the line associated with the speed of sound of the non-interacting quark gas) as a function of $|\Delta_0(n^*_{c_s})|$ (value of the gap at the crossing density). The red line is associated with the calculation employing a functional form of the gap as found in the weak-coupling limit, whereas the blue dashed line corresponds to the calculations employing a gap with a functional form as found in a recent fRG calculation, see main text for details. The dots are associated with the dashed vertical lines in Fig. 2.

In the spirit of our present study, this minimum may be used to divide strong-interaction matter into different regimes. For $n > n_{\mathrm{min}}$, we find that gap-induced corrections become subleading and may be dropped, not only in computations of the pressure but also in computations of the speed of sound. However, for $n < n_{\mathrm{min}}$, the gap leaves a clear imprint in the speed of sound. In fact, below $n_{\mathrm{min}}$, we find that gap-induced corrections lead to a qualitative change of the density dependence of the speed of sound.[5] To be more specific, we observe an increase of the speed of sound towards lower densities such that it eventually exceeds its asymptotic value at the "crossing density" $n^*_{c_s}$, i.e., the density at which the speed of sound crosses the line associated with the speed of sound of the non-interacting quark gas, see dots and vertical lines in Fig. 2. However, the actual value of this characteristic quantity depends on the density dependence of the gap and its size as measured by the parameter $\Delta_\star$, which is defined to be the size of the gap at $n/n_0 = 10$, $\Delta_\star = |\Delta_0(n/n_0 = 10)|$. In our calculations, we vary $\Delta_\star$ by a simple global rescaling of the gap with a constant parameter, see, e.g., Eq. (16).

In Fig. 3, we show the crossing density $n^*_{c_s}$ as a function of $|\Delta_0(n^*_{c_s})|$ which is the value of the gap at $n = n^*_{c_s}$. Loosely speaking, we observe that a larger value of the crossing density $n^*_{c_s}$ comes along with a larger value of the gap at the crossing density. Thus, for a large color-superconducting gap, we expect the speed of sound to exceed its asymptotic limit already at high densities. Interestingly, the crossing density $n^*_{c_s}$ is not only sensitive to the size of the gap. Our results suggest that this quantity is also (very) sensitive to the functional form of the

---

[5]The scale $n_{\mathrm{min}}$ only represents a rough estimate for the lower bound of the actual density above which gap-induced effects may be safely neglected in calculations of the speed of sound. It is not a rigorous scale on the basis of which the relevance of gap effects can be estimated in terms of power-counting arguments in the spirit of, e.g., effective field theories. Indeed, coming from high densities, gap effects can already be sizeable at $n = n_{\mathrm{min}}$ (see Fig. 2) but they have not yet changed the slope of the speed of sound as a function of the density.

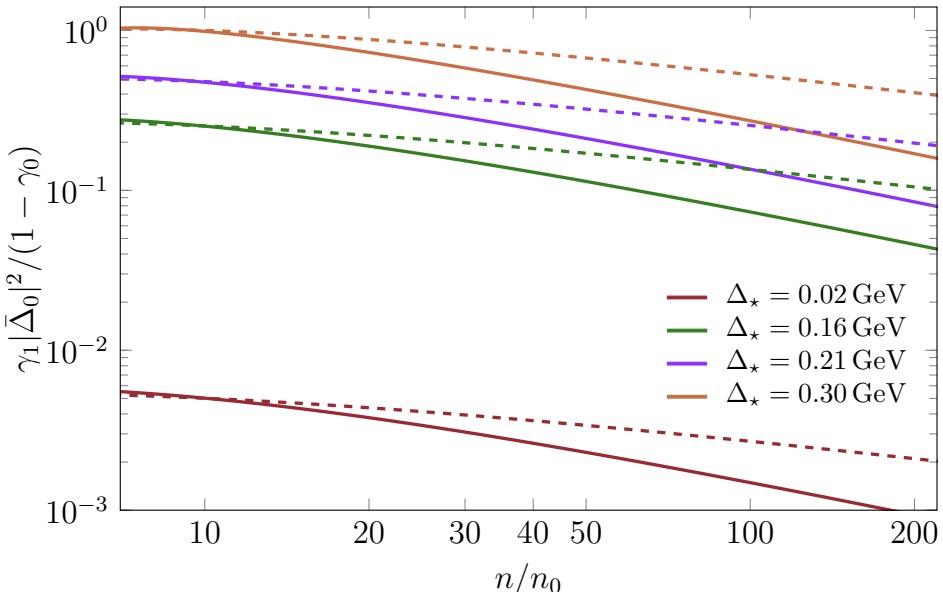

Figure 4: Size of the leading-order gap-induced correction to the pressure relative to the size of the leading-order perturbative correction. The solid lines are associated with calculations employing the functional form of the gap found in the weak-coupling limit whereas the dashed lines are associated with calculations employing the functional form of the gap found in a recent fRG study. As discussed in the main text, the gaps have been rescaled to estimate the effect of the size of the gap which as parametrized by the parameter $\Delta_\star$ (size of the gap at $n/n_0 = 10$).

gap, i.e., its density dependence. This can be deduced from a comparison of our results for a gap with a functional form as found in the weak-coupling limit with those for a gap with a functional form as found in a recent fRG study, see Figs. 2 and 3.

The existence of the crossing density can indeed be related to the properties of the color-superconducting gap, such as its size *and* its dependence on the chemical potential (or density). At least qualitatively, this can be seen by inserting the expansion (10) into the definition of the speed of sound, see Eq. (1). Taking into account only terms up to order $\sim |\bar{\Delta}_0|^2$, we find

$$c_s^2 = \frac{1}{3} + \frac{\pi^2}{6\mu^3} \frac{\partial}{\partial\mu} P_{SB} \left( \gamma_0 - 1 + \gamma_1 |\bar{\Delta}_0|^2 \right) + \dots \tag{17}$$

Here, we have assumed that corrections to the non-interacting quark gas are sufficiently small such that the denominator in Eq. (1) can be approximated by the expression for the non-interacting quark gas. From Eq. (17), we can read off that the speed of sound exceeds its asymptotic value, provided that

$$\frac{\partial}{\partial\mu} P_{SB} \gamma_1 |\bar{\Delta}_0|^2 > \frac{\partial}{\partial\mu} P_{SB} (1 - \gamma_0) . \tag{18}$$

For $\gamma_0 = 1$, it immediately follows that $c_s^2 > 1/3$, provided that $P_{SB}\gamma_1|\bar{\Delta}_0|^2$ increases with the chemical potential, see our discussion above and Ref. [48]. Using the result for $\gamma_0$ at leading order in the strong coupling, see Eq. (14), we find that

$$P_{SB}(1 - \gamma_0) \sim \frac{\mu^4}{\ln(\mu/\Lambda_{QCD})}, \tag{19}$$

which determines the $\mu$-dependence of the right-hand side of Eq. (18). The dependence of the left-hand side of Eq. (18) on the chemical potential depends on the $\mu$-dependence of the gap. In order to quantify the latter within the considered range of chemical potentials $(0.5\,\text{GeV} \lesssim \mu \lesssim 1.5\,\text{GeV})$, we assign an effective scaling exponent $\sigma$ to the gap, $|\Delta_0| \sim \mu^\sigma$. This leads us to

$$P_{\text{SB}}\gamma_1 |\bar{\Delta}_0|^2 \sim \mu^{2(1+\sigma)}\,. \tag{20}$$

A fit to the numerical data yields $\sigma \approx -0.19$ for the gap obtained in the weak-coupling limit and $\sigma \approx 0.16$ for the gap computed in the fRG study. From Eq. (17), we can now in principle obtain an estimate for $\mu_{c_s}^*$, i.e., the value of the chemical potential at which the speed of sound exceeds its asymptotic value.[6] In any case, these simple considerations already illustrate that $\mu_{c_s}^*$ (and therefore also the crossing density $n_{c_s}^*$) depends significantly on two properties of the color-superconducting gap: the functional form of the $\mu$-dependence of the gap as measured by its first derivative with respect to $\mu$ and the size of the gap which effectively appears as a constant of proportionality on the right-hand side of Eq. (20).

In Fig. 3, we indeed observe that the functional form of the gap clearly affects the value of the crossing density $n_{c_s}^*$. To be specific, for a given value of the crossing density $n_{c_s}^*$, the value of the gap at the crossing density is found to be (significantly) smaller in the calculations employing the functional form of the gap found in the weak-coupling limit than in the calculations employing the functional form of the gap found in the fRG study, at least for $n_{c_s}^* \lesssim 40$. However, we also observe that the functional form of the gap becomes less relevant in "scenarios" with large gaps.

The relevance of the functional form of the gap can also be illustrated by considering the pressure, see Eq. (12). To this end, we compare the leading-order gap-induced correction to the pressure of the non-interacting quark gas with the leading-order perturbative correction to the pressure of the non-interacting quark gas, see Fig. 4. Of course, the relevance of the gap-induced corrections to the pressure increases trivially with the size of the gap. However, in general, this does not necessarily entail a qualitative change of the density scaling of the speed of sound compared to calculations where gap-induced corrections are not taken into account. Also, comparatively small gap-induced corrections to the pressure may still lead to a qualitative change of the speed of sound as a function of the density, depending on the functional form of the gap. To be concrete, let us consider the green line associated with the crossing density $n_{c_s}^* \approx 8$ and $|\Delta_0(n_{c_s}^*)| \approx 0.16\,\text{GeV}$ in Fig. 3. For this green line, which has been computed by employing the functional form of the gap found in the weak-coupling limit, we find that the gap-induced corrections to the pressure are smaller than the perturbative corrections by a factor of four for $n \approx n_{c_s}^*$. At $n/n_0 \approx 28$ (where the corresponding speed of sound assumes a local minimum, see green line in the top panel of Fig. 2), the gap-induced corrections to the pressure are already smaller than the perturbative corrections by a factor of six.[7] Nevertheless, the gap-induced corrections lead to a qualitative change of the density

---

[6]Our considerations can also be used to obtain a simple estimate for the scaling behavior of the speed of sound above the crossing density (in a regime where the scaling of the gap can be described by the exponent $\sigma$). Indeed, assuming $\mu \sim n^{1/3}$, we find

$$c_s^2 = \frac{1}{3} + \bar{c}_0(1+\sigma)n^{\frac{2(\sigma-1)}{3}} - \frac{\bar{c}_1}{\ln(\bar{c}_2 n^{\frac{1}{3}})}\,,$$

where $\bar{c}_0, \bar{c}_1, \bar{c}_2$ are positive constants and $\bar{c}_2 n^{\frac{1}{3}} > 1$ within the validity range of this estimate. This "scaling law" illustrates the importance of the functional form of the gap (as measured by the exponent $\sigma$). Note that $\sigma < -1$ disfavors the emergence of a local minimum in the speed of sound as well as the appearance of a maximum with $c_s^2 > 1/3$ towards lower densities, at least at this order of the expansion. Moreover, by requiring that $|\Delta_0|/\mu \to 0$ for $\mu \to \infty$, it follows that $\sigma < 1$. We add that gap-induced corrections to the equation of state may potentially also exhibit a logarithmic scaling. For example, we may have $|\Delta_0| \sim \mu^{\bar{\sigma}} \ln \mu$ ($\bar{\sigma} < 1$). Then, the power-law scaling of the gap-induced term in the expression for $c_s^2$ is altered by a logarithmic correction.

[7]Note that, at such high densities, the perturbative correction $\sim g^2$ corresponds to $\sim 30\%$ of the pressure of the

dependence of the speed of sound. Thus, even if the gap-induced corrections to the pressure appear small, derivatives of the gap with respect to the chemical potential $\mu$ (as they enter the speed of sound) can still be sizeable because of its nontrivial dependence on $\mu$.

In this work, we restrict ourselves to densities where the ground state is not governed by spontaneous chiral symmetry breaking and the quarks remain massless. However, in principle, the quarks come with a finite current quark mass which we have also not included in our analysis. In addition to quark masses, which break the chiral symmetry, there are also dynamically generated in-medium self-energy corrections to the quarks, which do not break the chiral symmetry and come with a prefactor $\sim g^2\mu^2$ (at leading order in the coupling), see, e.g., Refs. [49–52]. Such self-energy corrections appear in the coefficients $\gamma_i$ of our expansion (10) of the equation of state. For example, quark self-energy corrections are taken into account in the perturbative two-loop result for the pressure which is included in the coefficient $\gamma_0$, see Eq. (14). The corresponding $g^2$-correction in the coefficient $\gamma_1$ will be presented elsewhere [73]. With respect to the relevance of our analysis for astrophysical applications, we also would like to briefly comment on the effect of a finite (constant) current quark mass $m_q$. First of all, we note that an inclusion of such a mass in our analysis would generate new terms in the expansion (10) of the equation of state. Indeed, the presence of a current quark mass potentially gives rise to terms $\sim m_q^2\mu^2$ and $\sim m_q^2|\Delta_0|^2$ in this expansion, which correspond to the gap-induced term $\sim \mu^2|\Delta_0|^2$. Here, we ignore higher-order terms in $m_q$ as we also dropped the corresponding higher-order terms in $|\Delta_0|$ in our considerations above. Assuming now that $m_q$ is small compared to the chemical potential *and* also small compared to the gap, we expect that $m_q$-induced corrections do not alter our general observations regarding the relevance of gap-induced corrections in the speed of sound, at least in the density range considered in this work.

We close by adding that our observations are in accordance with a non-perturbative study of the thermodynamics of dense strong-interaction matter as presented in Ref. [23], where RG flows starting from the QCD action are considered. There, the pressure computed in the presence of a gap at high densities is found to be consistent with the one from calculations which do not take into account a color-superconducting gap. However, towards lower densities, the presence of a gap has also been found to make a significant difference and eventually leads to an emergence of a maximum in the speed of sound which exceeds its asymptotic value. Interestingly, in that work, it was also observed that results from nonperturbative RG calculations, where the color-superconducting gap in the quark excitation spectrum has not been taken into account, do not exhibit an increase of the speed of sound above its value in the non-interacting limit when the density is decreased starting from (very) high densities. We close by noting that the existence of an increase of the speed of sound above the value associated with the non-interacting quark gas has also been observed in low-energy models, where QCD matter is studied coming from low densities (see, e.g., Refs. [74–78]) rather than from high densities as done in our work.

## 4 Conclusions

Our analysis of the speed of sound in dense strong-interaction matter with two massless quark flavors builds on an expansion of the equation of state in terms of the color-superconducting gap. We have discussed this expansion in detail. For example, we have pointed out that the zeroth-order term of this expansion can be directly related to the pressure as, e.g., computed in perturbative studies of dense strong-interaction matter. The first gap-induced term in this

---

non-interacting quarks. Since the gap-induced correction corresponds to $\sim 16\%$ of the perturbative correction, the gap-induced correction to the pressure corresponds to only $\sim 5\%$ of the pressure of the non-interacting quark gas.

expansion can be constrained by the fact that strong-interaction matter is expected to be a superconductor at sufficiently high densities. Gap-induced corrections of even higher order in this expansion appear to be parametrically suppressed at high densities.

Starting in the infinite-density limit, our analysis based on the aforementioned expansion of the pressure shows that the speed of sound first decreases, even in the presence of a color-superconducting gap. This observation is in agreement with first-principles studies of dense strong-interaction matter where such a gap in the excitation spectrum of the quarks has not been taken into account, see, e.g., Refs. [23, 53–58]. However, towards lower densities, the gap-induced corrections become increasingly important and lead to the emergence of a local minimum in the speed of sound at high densities. Above the density associated with this minimum, gap-induced corrections are small and our analysis even suggests that the corresponding contributions to the equation of state may be safely neglected in studies of thermodynamic properties of dense strong-interaction matter. Below the density associated with this minimum, gap-induced corrections to the equation of state become significant. In fact, these corrections induce an increase of the speed of sound when the density is further decreased such that the speed of sound eventually crosses the line associated with the speed of sound of the non-interacting quark gas. Taking into account results from studies based on chiral EFT interactions at low densities [22, 23], the existence of such a "crossing density" suggests the existence of a maximum in the speed of sound, in accordance with Ref. [23]. Of course, a quantitative determination of the position of this maximum is very challenging as the dynamics for $n/n_0 < 10$ is expected to be governed by a huge variety of interaction channels (including vector channels), which become equally relevant towards the nucleonic low-density regime [23], see also Refs. [74, 75, 79–86]. With respect to our expansion of the equation of state, we note that higher-order corrections become relevant in this low-density regime. In particular, the computation of the coefficients $\gamma_i$ may require non-perturbative methods.

Interestingly, we have found that the actual values of the crossing density and the density associated with the aforementioned local minimum in the speed of sound are not predominantly determined by the size of the gap but depend also significantly on the functional form of the gap (i.e., its dependence on the chemical potential). In particular, the value of the crossing density can be analytically related to the first derivative of the gap with respect to the chemical potential. Thus, even if the gap-induced contributions to the pressure may appear small, derivatives of the gap with respect to the chemical potential may be sizeable and therefore significantly affect the density dependence of the speed of sound. This observation is confirmed by our numerical studies.

With respect to astrophysical applications, it should be added that strange quarks may become relevant in the density regime considered in this work. In this regard, we would like to add that the mechanism, which pushes the speed of sound above its value in the non-interacting limit when the density is decreased, may be very general and not only a special feature of color superconductivity of the 2SC type discussed here. The same mechanism may also be at work if the gap in the quark excitation spectrum is of another type, such as the color-flavor locking (CFL) type in QCD with $2+1$ quark flavors. In fact, the expansion (5) of the pressure should assume the same functional form at least at leading order, i.e., the leading order gap-dependent correction should be also of the form $\sim \mu^2 |\Delta_0|^2$, where $|\Delta_0|$ now refers to, e.g., the CFL gap.[8] If the chemical potential dependence of this gap is similar to the one of the 2SC gap considered here, then also this gap potentially leads to an increase of the speed of sound above its value in the non-interacting limit, see also the discussion in the appendix of Ref. [48].

---

[8]From a thermodynamic standpoint, searching for the ground state corresponds to searching for the phase with highest pressure. Thus, if the ground state is governed by the presence of a gap, then the gap induces an increase of the pressure relative to the pressure in the absence of the gap.

With respect to the maximum in the speed of sound, we add that constraints from neutron-star masses also strongly support the existence of a global maximum in the speed of sound in neutron-rich matter [13–19]. In particular, the existence of such a maximum in the speed of sound and a local minimum at high densities has already been discussed in an analysis of constraints from astrophysical observations in Ref. [13]. It is also interesting to *speculate* whether it is possible to use constraints on the speed of sound from nuclear physics and observations to constrain the properties of color-superconducting matter. To be more specific, constraints on the speed of sound from observations provide estimates for lower bounds of the crossing density [17–19]. Since our present study suggests that this density can be related to the size of the gap, constraints on the crossing density allow to draw conclusions on the size of the gap in dense strong-interaction matter, see Fig. 3. For example, according to our present analysis, a crossing density of $8n_0$ requires that the color-superconducting gap assumes values of about 160 MeV in this density regime.

Of course, a quantitative computation of, e.g., the crossing density and the associated size of the gap requires the inclusion of higher-order corrections in our expansion of the pressure. However, this is beyond the scope of this work. Our present study rather aims at a better understanding of the mechanisms determining the density dependence of the speed of sound in dense strong-interaction matter. Still, we believe that our present analysis already adds to our understanding of the dynamics underlying strong-interaction matter at high densities.

## Acknowledgments

The authors would like to thank T. Gorda, K. Hebeler, J. M. Pawlowski, D. Rischke, and A. Schwenk for useful discussions. As members of the fQCD collaboration [87], the authors also would like to thank the other members of this collaboration for discussions.

**Funding information** J.B. acknowledges support by the Deutsche Forschungsgemeinschaft (DFG, German Research Foundation) under grant BR 4005/4-1 and BR 4005/6-1 (Heisenberg program). This work is supported by DFG – Projekt-ID 279384907 – SFB 1245 and by the State of Hesse within the Research Cluster ELEMENTS (Project No. 500/10.006).

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
