# Peer review of "Speed of sound in dense strong-interaction matter"

_SciPost Physics Core, doi:SciPost Phys. Core 7, 015 (2024)_

## Round 1 · Referee Report · Anonymous (Referee 1) · 2023-12-5

Report

The authors discuss the speed of sound at zero temperature in ultra-dense matter, in particular the effect of a color-superconducting gap. The full behavior of the speed of sound at nonzero chemical potential is unknown and has been discussed from various different angles in recent years. In particular, it is of relevance for the understanding of matter inside neutron stars. The manuscript adds an interesting and carefully discussed aspect to this area of research. The authors put together results from previous work in an original way and obtain interesting and novel results. The manuscript is mostly written in a clear and understandable way. I do have a few questions that I think the authors should address before publication:

(1) The authors neglect quark masses without really commenting on this approximation. Of course, the quark masses become negligibly small at high densities, but so does the color-superconducting gap. So is it clear that they would not change any of the conclusions? Since they will effectively be functions of the chemical potential it seems to me they could in principle have an effect on the speed of sound.

(2) I have a few questions about Fig 1: As mentioned in the text, the pQCD gap increases with mu (below Eq (13)); why is this not reflected in the weak-coupling curve in the upper panel? Also, from looking at the different scales for the 2 curves, it seems the fRG result does not approach the weak-coupling result at high densities. Is that correct? Maybe a comment and explanation would be helpful.

(3) I think it might be useful to mention that all results only hold for zero temperature already before the first paragraph of sec II. T=0 is already needed for Eq (1) and possibly it would even be useful for the reader to mention T=0 in the abstract.

(4) The authors argue that gap effects for n>n_min are negligible while they are important for n<n_min (page 6 and conclusions). This statement seems a bit imprecise to me since it is not based on any power counting, or is it? Looking at, say, the purple curve in the left panel of Fig 2, the deviation from the Delta=0 curve at n_min is already as large as the added deviation below n_min at the end of the scale. So I am not sure why n_min would play any role as a threshold for the power counting. Perhaps this statement can be made more precise.

(5) The authors rightly emphasize that the gap gives rise to a speed of sound larger than the asymptotic value and thus supports the expectation of a maximum in the speed of sound predicted on phenomenological grounds. Is the gap the only known physical effect that can yield this behavior in the high-density regime? For instance, if the quoted fRG study is repeated without the gap, would some pure interaction effects be able to produce a similar increase in the speed of sound? The result of the paper could be made even stronger if a short discussion would be added.

(6) In the conclusions, the authors mention that strange quarks are relevant for astrophysical applications. But even from a purely theoretical point of view, strange quarks (and all heavier flavours in principle) should appear at the densities considered here. The authors do mention earlier that 2+1 flavours are the "most relevant" case, but I would think a few more comments would be helpful here. For instance, at asymptotic densities we expect CFL rather than 2SC that is considered here. Would we expect any of the results to change qualitatively in that case?

---

## Round 2 · Author Response

* * *
(1) We agree that an at least brief discussion of the effect of the role of quark masses should be included in our manuscript. To this end, we distinguish between an explicit (constant) current quark mass $m_q$, which breaks the chiral symmetry explicitly, and in-medium self-energy corrections to the quarks, which do not break the chiral symmetry and come with a prefactor $\sim g^2 \mu^2$ (at leading order in the coupling).
With respect to the pressure, the current quark mass $m_q$ enters the equation of state via a term $\sim \mu^2 m_q^2$ (at lowest order in $m_q$). Moreover, a term $\sim m_q^2 |\Delta_0|^2$ is potentially present. Since the gap enters the equation of state as $\sim \mu^2 |\Delta_0|^2$ at leading order, we expect that the $m_q^2$-contributions to the equation of state (and also to the speed of sound) are subleading in the presence of the gap $\Delta_0$, provided that $m_q$ is small compared to the chemical potential and also small compared to the gap.
The situation is different in case of the aforementioned self-energy corrections to the quarks. These corrections already appear in the term $\sim\mu^4$ in the equation of state. In fact, to order $g^2$, they are implicitly taken into account in the two-loop result for the pressure which appears in our definition of the coefficient $\gamma_0$ in Eq. (14).
We have added a comment on these aspects on page 8 of the revised version (from line 26 of the left column to line 9 of the right column, "In this work, we restricted our analysis ... the density range considered in this work.").
* * *
(2)
a) We agree with the referee that our discussion is slightly misleading at this point. The gap computed in the weak-coupling limit increases for small chemical potentials and then decreases with increasing chemical potential, as shown in Fig. 1. Only at extremely large chemical potentials, way beyond any value which would be relevant for astrophysical applications, this gap increases again with increasing chemical potential. Note that $|\Delta_0|/\mu$ is still decreasing, even in this regime. In any case, in Fig. 1, we do not show the gaps for such extremely large values of the chemical potential.
For our general line of arguments, it is only necessary that $|\Delta_0|/\mu \to 0$ for large $\mu$, which is true for both "parametrizations" of the gap considered in our present work. To clarify this point, we have slightly modified the discussion below Eq. (13), see page 4 of the revised version (line 8-11 counted from Eq. (13), "Assuming that ... (i.e., in the limit of infinite density n).").
b) We have added a comment on the difference in the $\mu$-dependence of the gap as obtained from the weak-coupling calculation and the fRG study presented in Ref. [47], see the new footnote #4 on page 5/6. Please note that we have also included a new reference in this footnote, see Ref. [72] of the revised version of the manuscript.
* * *
(3) We agree with the referee. We now already state in the abstract that we only consider the zero-temperature limit (see first line of the abstract of the revised version of our manuscript). Moreover, we have rephrased the paragraph around Eq. (1) to make clear that only the zero-temperature limit is considered in our present work (see page 1 of the revised version, left column, line 12-18, "In the present work, ... the pressure of strong-interaction matter.").
* * *
(4) We did not intend to introduce the density $n_min$ as a strict scale on the basis of which power-counting arguments can be made. We only intended to introduce this scale as a rough estimate for the range of validity of calculations of the speed of sound in which the presence of a gap has not been taken into account. Coming from high densities, we agree with the referee that gap effects can already be sizeable at this scale but they have not yet changed the slope of the speed of sound as a function of the density. The density $n_min$ can therefore only represent a rough estimate for the lower bound of the actual density scale above which gap effects may be safely neglected in calculations of the speed of sound.
We have adapted our discussion of this aspect on page 6 to make clear that the scale $n_{\text{min}}$ only provides a rough estimate here. To be specific, we have included a new footnote on page 6 of the revised version (see new footnote #5) and slightly modified the wording of one sentence on page 6 (right column, line 7-11, "Importantly, we also observe ... speed of sound at $n = n_{\text{min}}$.").
* * *
(5) We cannot conclusively answer this question. However, in Ref. [23], the speed of sound has already been computed in two ways using the fRG approach, namely with and without gap, and it was observed that it requires the presence of a gap to push the speed of sound above its value in the non-interacting limit when the density is decreased, see right panel of Fig. 1 in Ref. [23].
We slightly modified our discussion at the end of Sec. III and added a comment on this aspect to the manuscript, see page 8 of the revised version of our manuscript (right column, line 21-33, "Interestingly, in that work, it was also observed that ... than from high densities as done in our work.").
* * *
(6) We expect that the presence of a color superconducting gap also leads to an increase of the speed of sound above its value in the non-interaction limit if color-flavor locking (CFL) is the dominant pairing mechanism in QCD with 2+1 quark flavors. In fact, in this case, the functional form of the expansion of the pressure given in Eq. (5) should remain unchanged (at least at the order given in this equation), i.e., the leading order gap-dependent correction should be of the form $\sim \mu^2 |\Delta_0|^2$, where $|\Delta_0|$ now refers to the CFL gap. If the dependence of this gap on the chemical potential is qualitatively similar to the ones discussed for the 2SC gap in the present work, then the gap also leads to an increase of the speed of sound above its value in the non-interaction limit in the case of color-flavor locking, see also the discussion in appendix B of Ref. [48].
We added a comment on this aspect in Sec. IV ("Conclusions"), see page 9 of the revised version of our manuscript (line 45 of the left column to line 9 of the right column, "In this regard, we would like to add ... above its value in the non-interacting limit, see also the discussion in the appendix of Ref. [48]."). Moreover, we reformulated the last sentence of Sec. I of the revised version (page 2, left column, line 9-10, "A brief discussion ... together with our conclusions.").
* * *
Additional changes:
(A) There was a typo in the label of the y-axis of Fig. 1. We corrected this typo and replaced the figure accordingly.
(B) We have updated Ref. [48].
(C) We have slightly modified the wording of two sentences in the Conclusions, see page 9 (right column, line 10-17, "With respect to the maximum ... astrophysical observations in Ref. [13].").
* * *
We hope that with these modifications the present version of our manuscript is now suitable for publication in SciPost.

---

## Round 2 · List of Changes

Jens Braun on 2024-02-13 [id 4310]
Warnings issued while processing user-supplied markup:
Add "#coerce:reST" or "#coerce:plain" as the first line of your text to force reStructuredText or no markup.
You may also contact the helpdesk if the formatting is incorrect and you are unable to edit your text.
(Remark: Because of a problem in the source text (associated with a markdown command) our initially submitted reply letter is displayed in an unreadable format in the field "Author comments upon resubmission". We fixed the problem with the markdown command in the source text and make our reply letter available here for the time being.)
We are grateful to the referee for their positive report and for stating that our manuscript adds an ``interesting and carefully discussed aspect'' regarding the properties of dense strong-interacting matter. Moreover, we thank the referee for their helpful comments and suggestions regarding the presentation of our results. Below, we reply to the referee's comments and suggestions following the order in which they appear in the report:
(1) We agree that an at least brief discussion of the effect of the role of quark masses should be included in our manuscript. To this end, we distinguish between an explicit (constant) current quark mass $m_q$, which breaks the chiral symmetry explicitly, and in-medium self-energy corrections to the quarks, which do not break the chiral symmetry and come with a prefactor $\sim g^2 \mu^2$ (at leading order in the coupling).
With respect to the pressure, the current quark mass $m_q$ enters the equation of state via a term $\sim \mu^2 m_q^2$ (at lowest order in $m_q$). Moreover, a term $\sim m_q^2 |\Delta_0|^2$ is potentially present. Since the gap enters the equation of state as $\sim \mu^2 |\Delta_0|^2$ at leading order, we expect that the $m_q^2$-contributions to the equation of state (and also to the speed of sound) are subleading in the presence of the gap $\Delta_0$, provided that $m_q$ is small compared to the chemical potential and also small compared to the gap.
The situation is different in case of the aforementioned self-energy corrections to the quarks. These corrections already appear in the term $\sim\mu^4$ in the equation of state. In fact, to order $g^2$, they are implicitly taken into account in the two-loop result for the pressure which appears in our definition of the coefficient $\gamma_0$ in Eq. (14).
We have added a comment on these aspects on page 8 of the revised version (from line 26 of the left column to line 9 of the right column, "In this work, we restricted our analysis ... the density range considered in this work.").
(2)
a) We agree with the referee that our discussion is slightly misleading at this point. The gap computed in the weak-coupling limit increases for small chemical potentials and then decreases with increasing chemical potential, as shown in Fig. 1. Only at extremely large chemical potentials, way beyond any value which would be relevant for astrophysical applications, this gap increases again with increasing chemical potential. Note that $|\Delta_0|/\mu$ is still decreasing, even in this regime. In any case, in Fig. 1, we do not show the gaps for such extremely large values of the chemical potential.
For our general line of arguments, it is only necessary that $|\Delta_0|/\mu \to 0$ for large $\mu$, which is true for both "parametrizations" of the gap considered in our present work. To clarify this point, we have slightly modified the discussion below Eq. (13), see page 4 of the revised version (line 8-11 counted from Eq. (13), "Assuming that ... (i.e., in the limit of infinite density n).").
b) We have added a comment on the difference in the $\mu$-dependence of the gap as obtained from the weak-coupling calculation and the fRG study presented in Ref. [47], see the new footnote No. 4 on page 5/6. Please note that we have also included a new reference in this footnote, see Ref. [72] of the revised version of the manuscript.
(3) We agree with the referee. We now already state in the abstract that we only consider the zero-temperature limit (see first line of the abstract of the revised version of our manuscript). Moreover, we have rephrased the paragraph around Eq. (1) to make clear that only the zero-temperature limit is considered in our present work (see page 1 of the revised version, left column, line 12-18, "In the present work, ... the pressure of strong-interaction matter.").
(4) We did not intend to introduce the density $n_\text{min}$ as a strict scale on the basis of which power-counting arguments can be made. We only intended to introduce this scale as a rough estimate for the range of validity of calculations of the speed of sound in which the presence of a gap has not been taken into account. Coming from high densities, we agree with the referee that gap effects can already be sizeable at this scale but they have not yet changed the slope of the speed of sound as a function of the density. The density $n_{\text{min}}$ can therefore only represent a rough estimate for the lower bound of the actual density scale above which gap effects may be safely neglected in calculations of the speed of sound.
We have adapted our discussion of this aspect on page 6 to make clear that the scale $n_{\text{min}}$ only provides a rough estimate here. To be specific, we have included a new footnote on page 6 of the revised version (see new footnote No. 5) and slightly modified the wording of one sentence on page 6 (right column, line 7-11, "Importantly, we also observe ... speed of sound at $n = n_{\text{min}}$.").
(5) We cannot conclusively answer this question. However, in Ref. [23], the speed of sound has already been computed in two ways using the fRG approach, namely with and without gap, and it was observed that it requires the presence of a gap to push the speed of sound above its value in the non-interacting limit when the density is decreased, see right panel of Fig. 1 in Ref. [23].
We slightly modified our discussion at the end of Sec. III and added a comment on this aspect to the manuscript, see page 8 of the revised version of our manuscript (right column, line 21-33, "Interestingly, in that work, it was also observed that ... than from high densities as done in our work.").
(6) We expect that the presence of a color superconducting gap also leads to an increase of the speed of sound above its value in the non-interaction limit if color-flavor locking (CFL) is the dominant pairing mechanism in QCD with 2+1 quark flavors. In fact, in this case, the functional form of the expansion of the pressure given in Eq. (5) should remain unchanged (at least at the order given in this equation), i.e., the leading order gap-dependent correction should be of the form $\sim \mu^2 |\Delta_0|^2$, where $|\Delta_0|$ now refers to the CFL gap. If the dependence of this gap on the chemical potential is qualitatively similar to the ones discussed for the 2SC gap in the present work, then the gap also leads to an increase of the speed of sound above its value in the non-interaction limit in the case of color-flavor locking, see also the discussion in appendix B of Ref. [48].
We added a comment on this aspect in Sec. IV ("Conclusions"), see page 9 of the revised version of our manuscript (line 45 of the left column to line 9 of the right column, "In this regard, we would like to add ... above its value in the non-interacting limit, see also the discussion in the appendix of Ref. [48]."). Moreover, we reformulated the last sentence of Sec. I of the revised version (page 2, left column, line 9-10, "A brief discussion ... together with our conclusions.").
Additional changes:
(A) There was a typo in the label of the y-axis of Fig. 1. We corrected this typo and replaced the figure accordingly.
(B) We have updated Ref. [48].
(C) We have slightly modified the wording of two sentences in the Conclusions, see page 9 (right column, line 10-17, "With respect to the maximum ... astrophysical observations in Ref. [13].").
We hope that with these modifications the present version of our manuscript is now suitable for publication in SciPost.

---

## Editorial Decision

published